# Phytochemicals and Antioxidant Capacities of Young Citrus Fruits Cultivated in China

**DOI:** 10.3390/molecules27165185

**Published:** 2022-08-15

**Authors:** Haitian Fang, Huiling Zhang, Xiaobo Wei, Xingqian Ye, Jinhu Tian

**Affiliations:** 1Ningxia Key Laboratory for Food Microbial Applications Technology and Safety Control, Ningxia University, Yinchuan 750021, China; 2College of Biosystems Engineering and Food Science, National-Local Joint Engineering Laboratory of Intelligent Food Technology and Equipment, Zhejiang Key Laboratory for Agro-Food Processing, Integrated Research Base of Southern Fruit and Vegetable Preservation Technology, Zhejiang International Scientific and Technological Cooperation Base of Health Food Manufacturing and Quality Control, Zhejiang University, Hangzhou 310058, China; 3Fuli Institute of Food Science, Zhejiang University, Hangzhou 310058, China; 4Ningbo Research Institute, Zhejiang University, Ningbo 315100, China; 5Food Health & Testing center, Zhejiang University Zhongyuan Institute, Zhengzhou 450000, China

**Keywords:** young citrus, phenolic acids, flavonoids, antioxidant activity, correlation

## Abstract

Fruits of six varieties of young citrus cultivated in China were collected for phytochemical composition analysis and antioxidant activity determination. The phenolic acids, synephrine, flavone, and flavanone were analyzed using HPLC, and the total phenolic content and antioxidant capacity were determined by Folin-Ciocalteu, Ferric ion reducing antioxidant power (FRAP), 2,2- 1,1-diphenyl-2-picrylhydrazyl (DPPH), and 2,2′-azino-bis-3-ethylbenzthiazoline-6-sulphonic acid (ABTS) analysis. The results indicated that Ougan variety had the highest total phenolic content (125.18 GAE mg/g DW), followed by the Huyou variety (107.33 mg/g DW), while Wanshuwenzhoumigan variety had the lowest (35.91 mg/g DW). Ferulic acid was the most dominant soluble phenolic acid in the selected young citrus, followed by *p*-coumaric acid and *p*-hydroxybenzoic acid, whereas nobiletin and tangeretin were the most abundant flavones in the Ponkan, Ougan, and Wanshuwenzhoumigan varieties. Antioxidant capacity that measured by ABTS, FRAP, and DPPH showed similar trends and was positively correlated with the total phenolic and total flavonoid contents (*p* < 0.05). Considering the high content of phenolics in the young fruits of Ougan and Huyou variety, those two varieties might be potential resources for extracting phytochemicals for health promotion.

## 1. Introduction

Phytochemicals are a large group of secondary metabolites found in vegetables and fruits, and they can be classified as carotenoids, phenolics, alkaloids, and/or organosulfur compounds, among others, depending on the variations of their chemical structures [1]. Numerous studies have demonstrated that consumption of a large amount of plant food rich in phytochemicals was negatively associated with the risks of chronic and degenerative diseases [2]. Citrus is one of the world’s major horticultural crops, with a global production of 100 million metric tons per year. In addition to the high content of sucrose, pectin and vitamin C, citrus fruits are also abundant in phytochemicals, such as phenolic acids, flavonoids, and synephrine, and are considered as a potential foodstuff to improve human health. For instance, the flavones from citrus fruits were regarded as a good resource in the regulation of antioxidant activity, anti-inflammatory, anti-tumor proliferation, while the synephrine from citrus fruits were normally considered to be useful in the body weight management as well as the cardiovascular disease relieving.

Up to now, many studies have focused on the identification and quantification of the phytochemicals and antioxidant capacity of citrus from different varieties (e.g., mandarin, grapefruit, sweet orange, lemon, and pummelo) or origins [3]. However, only a few studies have sought to determine the phytochemicals and antioxidant activity of young citrus fruits. Those fruits, which are also known as immature citrus fruits, have been used as traditional Chinese medicines (Zhi-Shi or Zhi-Ke) for thousands of years [4]. Recent studies have identified many biological constituents in young citrus fruits, such as flavonoids, phenolic acids, limonoids, and adrenergic amines (mainly synephrine) [5], and have claimed that young citrus could be used as a raw material source for flavonoid and synephrine production. However, to the best of our knowledge, only a few studies have focused on the composition and antioxidant activity of the bioactive compounds in those young fruits of different citrus varieties. Therefore, the objective of the present study was to compare the composition and distribution of total phenolic content, flavonoids, phenolic acids, synephrine, etc. as well as the antioxidant capacities in young citrus fruits from different varieties that commonly cultivated in China. The information would be useful for choosing phytochemical-extracting materials from citrus samples and improving the values of young citrus fruits.

## 2. Results

### 2.1. Total Phenolic, Total Flavonoid, and Synephrine Contents

The total phenolic, total flavonoid, and synephrine contents of young citrus varieties are showed in Figure 1. The values varied largely among citrus varieties, where Ougan variety achieved the highest total phenolic content (125.18 GAE mg/g DW), followed by Huyou variety (107.33 mg/g DW) and Wanshuwenzhoumigan variety (35.91 mg/g DW). Our results were in accordance with previous studies [6], who also reported a higher total phenolic content in Huyou variety than in Ponkan variety. The total flavonoid contents varied from 30.26 to 64.06 mg/g DW (rutin equivalents) and were much higher than those obtained by other researchers (4.671–5.796 mg rutin equivalents/g DW) [7] and (7.95–20.66 mg/g DW rutin equivalents) [8].

The difference in total flavonoid content between our study and others might attribute to the variety diversity, origin, or analysis methods. In the present study, the highest total flavonoid content was found in the Ougan variety (64.04 mg/g DW), followed by Huyou variety (49.63 mg/g DW), whereas the Tiancheng variety accounted for the lowest (30.26 mg/g DW). It seems that mandarin fruits had much higher content of synephrine than that of sweet orange and grapefruit. The highest synephrine content was found in the Wanshuwenzhoumigan variety (23.65 mg/g DW), followed by the Ougan variety (22.46 mg/g DW), Ponkan variety (18.99 mg/g DW), and Zaoshuwenzhoumigan variety (17.79 mg/g DW), while the Tiancheng and Huyou varieties accounted for about 12.03 and 3.01 mg/g DW, respectively. Our results were similar to some previous reports. Previous researchers reported the synephrine content of five species of citrus in Brazil ranging from 0.03 to 1.97 mg/g DW [9]. While in another study, researchers reported a range from 7.6 to 22.5 mg/g DW of synephrine in nine citrus varieties in China [5].

### 2.2. Soluble Phenolic Acids

Seven phenolic acids (caffeic, *p*-coumaric, ferulic, sinapic, protocatechuic, *p*-hydroxybenzoic, and vanillic) were measured by HPLC-PDA (see Appendix A), and the results are shown in Table 1. Ferulic acid was the most dominant extractable phenolic acid, followed by *p*-coumaric acid and *p*-hydroxybenzoic acid. The contents of these phenolic acids varied from 1804.28 to 7104.73 µg/g DW, 774.87 to 995.18 µg/g DW, and 120.71 to 359.82 µg/g DW in the selected young citrus fruits, respectively. The highest ferulic acid was observed in Ougan variety, while the highest *p*-coumaric acid content was found in Zaoshuwenzhoumigan variety and the highest *p*-hydroxybenzoic acid content was found in Huyou variety. The total phenolic acids were also calculated, and results indicated that the phenolic acids ranged from 3371.59 to 8604.17 µg/g DW. Our results were in accordance with Zhang et al. (2014) [8], but much lower than those of Ye et al. (2007) [5]. Those differences might attribute to the genetic backgrounds of the citrus species, environmental factors, and/or analysis methods.

### 2.3. Contents of Flavone and Flavanone

#### 2.3.1. Flavone

Seven major flavones (rhoifolin, quercitrin, luteolin, diosmetin, sinensetin, nobiletin, and tangeretin) of citrus fruits were determined, and the results are shown in Table 2. The main flavone compounds varied significantly among citrus varieties. Nobiletin and tangeretin were the most abundant compounds in the Ponkan, Ougan, and Wanshuwenzhoumigan varieties. The contents of these flavones varied from 0.31 to 11.9 mg/g DW and 0.17 to 10.21 mg/g DW, respectively. Rhoifolin and nobiletin were the major compounds in Huyou variety, sinensetin and nobiletin were the major compounds in Tiancheng variety, and diosmetin and nobiletin were the major compounds in Zaoshuwenzhoumigan variety. Our results were in accordance with previous researchers, who also found high contents of flavone compounds in different citrus species. In the present study, we also found that the nobiletin and tangeretin contents of Ougan variety were higher than those reported by other researchers [7,10], indicating that the young Ougan fruits used in this study might be an attractive material for extracting nobiletin and tangeretin.

#### 2.3.2. Flavanone

The flavanone compounds (eriocitrin, taxifolin, narirutin, naringin, hesperidin, neohesperidin, eridictyol, didymin, poncirin, and naringenin) in young citrus were also measured with HPLC, and the results are presented in Table 2. Narirutin and hesperidin were considered major flavanone compounds in mandarin (except Ougan, in which hesperidin and neohesperidin were the main compounds) and sweet orange. The narirutin ranged from 3.21 (Tiancheng variety) to 27.45 mg/g DW (Zaoshuwenzhoumigan variety), and hesperidin ranged from 5.38 (Ponkan variety) to 7.04 mg/g DW (Zaoshuwenzhoumigan variety). Naringin and neohesperidin were the main compounds in Huyou and accounted for about 13.29 and 11.65 mg/g DW, respectively. Eridictyol was not detectable in all of the selected young citrus, and naringenin was detected only in the Tiancheng variety at a very low concentration (0.07 mg/g DW). Our results were comparable with previous reports [11,12]. The total flavanones were also calculated, and the highest content was observed in Huyou variety (68.53 mg/g DW), followed by the Ougan variety (54.96 mg/g DW), while the lowest was observed in the Tiancheng variety (7.67 mg/g DW). This result revealed that the flavanone content in the Huyou and Ougan varieties could be a source for further utilization.

### 2.4. Antioxidant Capacity

The antioxidant activity of young citrus fruits was assessed by ABTS, FRAP, and DPPH assays, and the results were showed in Figure 2. The DPPH values of the selected young citrus varied from 91.67 to 170.58 μmoL TE/g DW. The Ougan variety had the highest DPPH value, followed by Wanshuwenzhoumigan variety. On the other hand, Tiancheng variety had the lowest DPPH value. The FRAP values of the six varieties of young citrus ranged from 118.71 to 723.11 μmoL TE/g DW. The highest FRAP value was found in Ougan variety, whereas the lowest was found in Ponkan variety. Considering the different values of antioxidant capacity calculated from DPPH, ABTS, and FRAP for the same young citrus, an overall antioxidant potency composite (APC) index were performed [13]. As showed in Figure 2D, the APC index of different young citrus ranged from 34.23 to 97.03 (Figure 2). The Ougan variety had the highest APC index, followed by the Huyou variety and Wanshuwenzhoumigan variety, whereas the Zaoshuwenzhoumigan variety and Ponkan variety had the lowest APC index.

### 2.5. Correlation between Antioxidant Activity and Selected Phytochemicals

Phytochemicals, including flavonoids, phenolic acids, and others, were known to be responsible for antioxidant activity in fruits. Thus, in order to gain a better understanding of the relationship between selected phytochemicals and antioxidant activity in young citrus, the correlations among total phenolic, total flavonoid, synephrine, ABTS, DPPH, and FRAP were analyzed; the results are presented in Table 3. Both total phenolic and total flavonoid content showed positive correlations with ABTS, FRAP, and DPPH and indicated that a higher total phenolic and/or total flavonoid content was responsible for the higher antioxidant activity. On the contrary, the correlation between synephrine and antioxidant activity was not significant and indicated that synephrine might not contribute much to antioxidant activity.

Chemical antioxidant activity assays (ABTS, FRAP, and DPPH) have been widely used to evaluate the antioxidant activity of citrus samples, and the mechanisms of those methods were based primarily on assessing hydrogen atom transfer and electron transfer during reaction [14]. In present study, the values of DPPH were positively correlated with FRAP and ABTS at an extreme level (*p* < 0.01), while values from FRAP and ABTS also showed a positive correlation at the 0.05 level. Thus, the results from different methods in the present study were comparable. In a previous study, the authors also reported a positive correlation among the values of FRAP, DPPH, and ABTS in the extracts of Chinese bayberry [15].

## 3. Materials and Methods

### 3.1. Samples

Six typical citrus varieties cultivated in Zhejiang Province, China, were selected for this research (each were collected about 10 kg). They were collected at a young stage from an orchard in Quzhou City, Zhejiang, China in May, 2015 (those young fruits dropped naturally; detailed information is listed in Table 4). The whole fruits of young citrus were dried in an oven at 50 ℃ for about 48 h, mashed with a grinder, and sieved through 200 meshes. Then, 1.000g powder samples were weighted accurately and mixed with 25 mL of 80% methanol in a brown glass tube, and the phytochemicals were ultrasound-assisted (950 W) extracted at 35 ℃ for about 40 min. The extracts were filtered through a 0.44 μm membrane and stored at 4 ℃ for future analysis.

### 3.2. Chemicals

The standards of phenolic acids (protocatechuic acid, *p*-hydroxybenzoic acid, vanillic, caffeic acid, *p*-coumaric acid, ferulic acid, sinapic acid), flavanone (eriocitrin, taxifolin, narirutin, neohesperidin, eridictyol, didymin, rhoifolin, poncirin, naringenin), flavone (quercitrin, luteolin, diosmetin, sinensetin, nobiletin, tangeretin), synephrine with chromatography grade and gallic acid, rutin, and Folin-Ciocalteu reagent with analytic grade were purchased from Sigma Aldrich Ltd. (St. Louis, MO, USA). Other chemicals and reagents used in the present study were purchased from Aladdin (Shanghai, China) and Sinopharm Chemical Reagent Co., Ltd. (Shanghai, China).

### 3.3. Determination of Total Phenolic, Total Flavonoid, and Synephrine Contents

#### 3.3.1. Total Phenolic Content

Total phenolic content was analyzed with Folin-Ciocalteu method according to previous studies [16]. Briefly, a certain volume of extracts was diluted, adjusted to 10 mL with distilled water, and mixed with 1 mL of Folin-Ciocalteu reagent; the mixture was kept in darkness for 5 min. After 5 mL of sodium carbonate (5%) was added, the volume was adjusted to 25 mL with distilled water and mixed well, and then the mixtures were incubated for 60 min at room temperature. The absorbance was measured at 765 nm (UV-2550, Shimadzu, Japan) with a standard curve of gallic acid.

#### 3.3.2. Total Flavonoid Content

Total flavonoid was measured according to other researchers with minor modification [17]. Briefly, a certain volume of extracts (0.5 mL) was transferred to a brown glass tube and adjusted to 5 mL with 30% ethanol and mixed with 0.3 mL of aluminum nitrate solution (10%). Then, 4 mL of 1 M NaOH were added, and the volumes of the mixture were adjusted to 10 mL with 30% ethanol. After being kept in the dark for about 10 min, the absorbance was measured at 510 nm (UV-2550, Shimadzu, Japan) with a standard curve of Rutin.

#### 3.3.3. Synephrine Content

The synephrine content in young citrus was analyzed with a Waters model 2995 separation system (Waters, Corp., Milford, MA, USA) using an Inertsil ODS-3 column (4.6 mm × 250 mm, 5 μm) [18]. An isocratic elution of distilled water (eluent A, containing 0.06% phosphoric acid and 0.1% sodium dodecyl sulfate) and methanol (eluent B) was applied at a ratio of 3:7. The flow rate was set at 0.8 mL/min with an injection of 5 μL samples, and synephrine was detected with a UV-visible photodiode array detector at a wavelength of 225 nm.

### 3.4. Measurement of Flavone and Flavanone

The contents of flavone and flavanone were measured according to a previous report [19]. An HPLC system (as described in Section 3.3.3) with a C18 reversed-phase column (Agilent Zorbax SB-C18; 250 mm × 4.6 mm, 5 μm) and a gradient elution of 0.1% formic acid in water (eluent A) and methanol (eluent B) was applied. The gradient increased from 37% to 40% B in the first 20 min, reached 80% B at 35 min, increased to 100% at 40 min, and maintained this level to 50 min and then decreased to 37% B at 60 min. The flow rate was set to 0.7 mL/min with a re-equilibration time of 42 min. Flavone and flavanone were detected with a UV-visible photodiode array (Waters model 2696) at a wavelength of 283 nm with an injection of 5 μL extracts (for eriocitrin, taxifolin, narirutin, neohesperidin, eridictyol, didymin, poncirin, and naringenin) and 330 nm (for rhoifolin, quercitrin, luteolin, diosmetin, sinensetin, nobiletin, and tangeretin), respectively.

### 3.5. Determination of Soluble Phenolic Acids

The soluble phenolic acids in young citrus were measured based on a previously described method [7]. Briefly, 10 mL of extracts were concentrated to 2 mL under reducing pressure at 45 ℃ and mixed with 4 M NaOH in a brown tube and maintained for about 8 h. Then, the pH was adjusted to 2 by adding 6 M HCl and shaking for about 2.5 h. The acid hydrolysis solutions were mixed with an extract solvent (ethyl acetate: diethyl ether, *v*/*v* 1:1) and filtered with filter paper. Then, the extracts were concentrated, the residues were dissolved in 5 mL methanol, and the ultrasonic was applied to promote dissolution. Finally, the solution was filtered through a 0.22 μm organic membrane and stored at 4 °C for future analysis.

The analysis of soluble phenolic acids was performed via an HPLC system with a C18 reversed-phase column (Agilent Zorbax SB-C18; 250 mm × 4.6 mm, 5 μm). An isocratic elution of 4% acetic acid (eluent A) and methanol (eluent B) was applied at a ratio of 2:8. The flow rate was set at 1 mL/min, with a re-equilibration time of 13 min. The injection of the extract was 5 μL and the soluble phenolic acids were detected with a UV-visible photodiode array detector at a wavelength of 260 nm (protocatechuic acid, *p*-hydroxybenzoic acid, and vanillic acid) and 320 nm (caffeic acid, *p*-coumaric acid, ferulic acid, and sinapic acid).

### 3.6. Antioxidant Activity Assay

#### 3.6.1. ABTS Assay

The ABTS assay was performed according to early studies with some modification [15]. Briefly, the extracts were diluted to an appropriate concentration with 80% methanol (for Ougan and Huyou were diluted 26 times, while others were 13.5 times), and then 0.1-mL samples were mixed with ABTS+ solution (3.9 mL) in a vortex, reacted in darkness for 10 min, and measured at 734 nm with a spectrophotometer (UV-1650PC, Shimadzu). A control containing 100 μL of methanol (no extract) was also mixed with the ABTS+ solution. Results were expressed as mM of Trolox equivalents (TE) per g of DW.

#### 3.6.2. FRAP Assay

The FRAP assay was performed based on a previous report [19]. Briefly, 10 mM of TPTZ (dissolved in 40 mM HCl), 20 mM of ferric chloride, and 0.1 mol/L of acetate buffer (pH 3.6) were mixed (1:1:10; *v*/*v*/*v*) to obtain a FRAP solution. One hundred μL samples (diluted to an appropriate concentration; for Ougan and Huyou were diluted 20 times, while others were 5 times) were mixed with 4.9 mL of FRAP solution and kept in darkness for about 10 min, and the absorbance was measured at 593 nm. A control containing 100 μL of methanol was also mixed with FRAP solution, and the results were expressed as mM of TE per g of DW.

#### 3.6.3. DPPH Assay

The DPPH assay was performed as previously described [20], with a slight modification. Briefly, 2.8 mL of DPPH solution (0.1 mM, dissolved in methanol) was mixed with 0.2 mL of diluted extracts (equal volume of 80% methanol was used as the control) and mixed in a vortex. The reaction for scavenging DPPH radicals was performed in darkness at room temperature and absorbance was recorded at 517 nm. Results were expressed as μmoL of TE per g of DW.

### 3.7. Statistical Analyses

All of the samples were performed in triplicates, and data were expressed as mean ± standard deviation (SD). Statistical analyses were performed using Microsoft Excel 2010 and SPSS (version 20.0; USA). Means were compared by Duncan’s new multiple range test. Statistically significant differences and extreme differences were set at *p* < 0.05 and *p* < 0.01, respectively.

## 4. Conclusions

The composition of phytochemicals and antioxidant capacities of young citrus were analyzed, and significant variations in total phenolic and total flavonoid contents were observed among those citrus varieties. The total phenolics were, in order from high to low, Ougan > Huyou > Tiancheng > Zaoshuwenzhoumigan > Ponkan > Wanshuwenzhoumigan. The order of total flavonoids was Ougan > Huyou > Wanshuwenzhoumigan > Zaoshuwenzhoumigan > Ponkan > Tiancheng. We also found that the young citrus was rich in phenolic acids, flavone, and flavanone. The content of ferulic acid was the highest in the selected young citrus fruits, followed by *p*-coumaric acid and *p*-hydroxybenzoic acid, whereas nobiletin and tangeretin were the highest flavones in Ponkan variety, Ougan variety, and Wanshuwenzhoumigan variety. Indicating that those young fruits were protentional materials to obtain targeted phytochemicals (e.g., phenolics, flavonoids). Additionally, the antioxidant capacities of those young fruits were measured with ABTS, FRAP, and DPPH, and different methods showed similar trends and were closely associated with the total phenolic and total flavonoid contents (*p* < 0.05). The APC analysis indicated that Ougan variety had the highest APC index, followed by Huyou variety. For the synephrine, which showed a similar structure with ephedrine, was considered as a protentional chemical in the management of body weight, showed high content in Ponkan variety, Ougan variety, Zaoshuwenzhoumigan variety and Wanshuwenzhoumigan. This varitiy might be a good resource for the extracting of synephrine. In general, young fruits of Ougan variety and Huyou variety were potential resources for extracting bioactive compounds for their high contents of phytochemicals, and our results provide useful information for the future study and utilization of the young citrus fruits of those varieties.

## Figures and Tables

**Figure 1 molecules-27-05185-f001:**
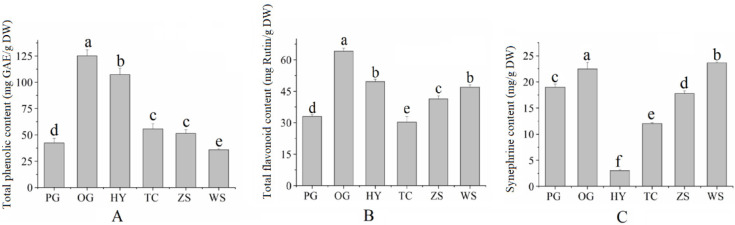
Total phenolic (**A**), total flavonoid (**B**) and synephrine content (**C**) in young citrus (PG: Ponkan; OG: Ougan; HY;Huyou; TC: Tiancheng; ZS: Zaoshuwenzhoumigan; WS: Wanshuwenzhoumigan; Different letter represent the significant difference at *p* < 0.05 level).

**Figure 2 molecules-27-05185-f002:**
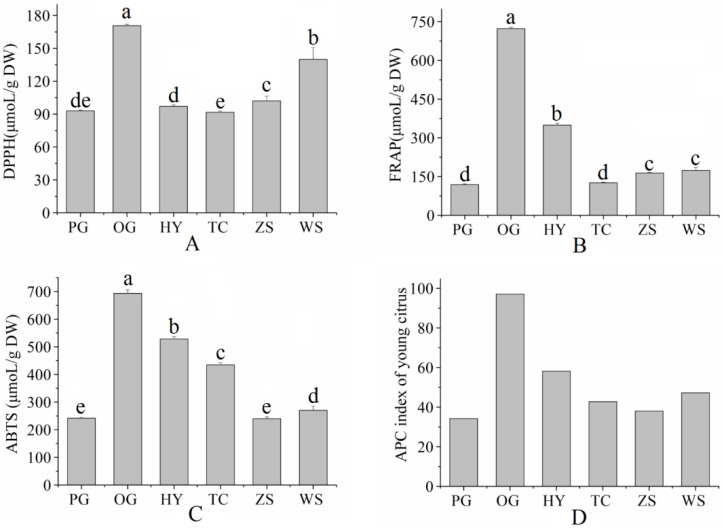
Antioxidant activity determined by DPPH (**A**), FRAP (**B**), ABTS (**C**) and their APC indexes (**D**) of young citrus (PG: Ponkan; OG: Ougan; HY;Huyou; TC: Tiancheng; ZS: Zaoshuwenzhoumigan; WS: Wanshuwenzhoumigan; a–e in each row indicated significant difference at 0.05 levels).

**Table 1 molecules-27-05185-t001:** Soluble phenolic acids contents in young fruits of citrus from different varieties (µg/g DW).

	PG	OG	HY	TC	ZS	WS
Protocatechuic acid	16.16 ± 1.87 ^b^	5.88 ± 0.24 ^d^	10.71 ± 6.53 ^bcd^	5.61 ± 1.47 ^d^	71.38 ± 22.52 ^a^	8.92 ± 0.30 ^c^
*p*-hydroxybenzoic acid	259.07 ± 3.51 ^b^	120.71 ± 0.83 ^c^	359.82 ± 17.26 ^a^	203.00 ± 72.13 ^b^	198.73 ± 4.39 ^b^	242.07 ± 10.03 ^b^
vanillic	107.42 ± 3.68 ^d^	39.49 ± 1.26 ^e^	381.55 ± 18.72 ^a^	137.15 ± 49.20 ^bcd^	155.73 ± 7.86 ^c^	178.03 ± 3.401 ^b^
caffeic acid	23.27 ± 0.02 ^d^	352.31 ± 81.76 ^a^	17.86 ± 1.97 ^e^	20.78 ± 1.39 ^de^	81.69 ± 14.37 ^b^	26.47 ± 0.16 ^c^
*p*-coumaric acid	979.93 ± 23.58 ^a^	937.09 ± 16.01 ^a^	739.12 ± 34.86 ^b^	774.87 ± 27.07 ^b^	995.18 ± 96.88 ^a^	952.96 ± 9.93 ^a^
ferulic acid	3524.80 ± 54.5 ^c^	7104.73 ± 104.59 ^a^	1804.28 ± 76.76 ^e^	3328.31 ± 11.71 ^d^	4279.97 ± 42.25 ^b^	4196.36 ± 55.96 ^b^
sinapic acid	42.78 ± 0.96 ^c^	43.94 ± 1.88 ^bc^	58.25 ± 2.14 ^a^	56.33 ± 9.68 ^ab^	65.67 ± 6.05 ^a^	48.66 ± 1.93 ^b^
Total	4953.44 ± 75.44 ^c^	8604.17 ± 43.03 ^a^	3371.59 ± 58.24 ^e^	4526.06 ± 58.87 ^d^	5848.34 ± 54.58 ^b^	5653.48 ± 54.24 ^b^

Note: a–e in each row indicated significant difference at 0.05 levels; (PG: Ponkan; OG: Ougan; HY; Huyou; TC: Tiancheng; ZS: Zaoshuwenzhoumigan; WS: Wanshuwenzhoumigan).

**Table 2 molecules-27-05185-t002:** Flavanone and flavone contents in young fruits of citrus (mg/g DW).

Genotypes		PG	OG	HY	TC	ZS	WS
	Eriocitrin	0.58 ± 0.02 ^d^	0.27 ± 0.02	1.43 ± 0.02 ^a^	0.88 ± 0.00 ^c^	0.62 ± 0.01 ^d^	1.15 ± 0.00 ^b^
	Taxifolin	ND	ND	0.13 ± 0.02 ^b^	0.46 ± 0.01 ^a^	0.11 ± 0.00 ^c^	0.08 ± 0.00 ^d^
	Narirutin	27.45 ± 0.38 ^b^	0.26 ± 0.02 ^f^	12.90 ± 0.69 ^d^	3.21 ± 0.02 ^e^	37.41 ± 0.49 ^a^	26.49 ± 0.21 ^c^
	Naringin	ND	15.33 ± 1.02 ^a^	13.29 ± 1.00 ^b^	0.23 ± 0.01 ^d^	0.10 ± 0.01 ^e^	0.31 ± 0.01 ^c^
Flavanone	Hesperidin	5.38 ± 0.43 ^d^	16.21 ± 1.08 ^a^	14.35 ± 0.18 ^b^	1.81 ± 0.03 ^e^	7.04 ± 0.07 ^c^	5.79 ± 0.05 ^d^
	Neohesperidin	1.15 ± 0.01 ^d^	32.51 ± 21.66 ^a^	11.65 ± 0.80 ^b^	0.54 ± 0.03 ^e^	0.98 ± 0.01 ^d^	3.02 ± 0.06 ^c^
	Eridictyol	ND	ND	ND	ND	ND	ND
	Didymin	1.47 ± 0.04 ^a^	0.28 ± 0.02 ^e^	0.52 ± 0.12 ^d^	0.38 ± 0.03 ^d^	2.05 ± 0.01 ^c^	1.12 ± 0.00 ^b^
	Poncirin	ND	3.67 ± 0.24 ^a^	0.69 ± 0.01 ^b^	0.16 ± 0.00 ^c^	ND	ND
	Naringenin	ND	ND	ND	0.07 ± 0.00	ND	ND
	rhoifolin	ND	0.24 ± 0.02 ^b^	1.71 ± 0.02 ^a^	0.16 ± 0.01 ^c^	ND	ND
	quercitrin	ND	1.61 ± 0.11 ^a^	0.49 ± 0.01 ^b^	ND	ND	ND
	luteolin	ND	0.03 ± 0.00	0.13 ± 0.00	0.04 ± 0.00	ND	0.02 ± 0.00
Flavone	diosmetin	0.18 ± 0.01 ^c^	ND	ND	ND	0.29 ± 0.01 ^b^	0.34 ± 0.00 ^a^
	sinensetin	ND	1.61 ± 0.11 ^a^	ND	1.30 ± 0.00 ^b^	0.05 ± 0.00 ^d^	0.10 ± 0.00 ^c^
	nobiletin	0.31 ± 0.01 ^f^	11.90 ± 0.79 ^a^	1.17 ± 0.01 ^c^	2.67 ± 0.00 ^b^	0.46 ± 0.01 ^e^	0.98 ± 0.01 ^d^
	tangeretin	0.17 ± 0.03 ^e^	10.21 ± 0.68 ^a^	0.60 ± 0.01 ^b^	0.28 ± 0.00 ^d^	0.18 ± 0.00 ^e^	0.48 ± 0.00 ^c^

Note: a–f in each row indicated significant difference at 0.05 levels, ND: not detected; (PG: Ponkan; OG: Ougan; HY;Huyou; TC: Tiancheng; ZS: Zaoshuwenzhoumigan; WS: Wanshuwenzhoumigan).

**Table 3 molecules-27-05185-t003:** Correlation analysis of antioxidant abilities in young fruits of citrus.

	Total Phenolic	Total Flavonoid	Synephrine	ABTS	DPPH	FRAP
Total phenolic	1.000					
Total flavonoid	0.721 **	1.000				
Synephrine	−0.029	0.036	1.000			
ABTS	0.956 **	0.633 **	−0.121	1.000		
DPPH	0.627 **	0.924 **	0.194	0.523 *	1.000	
FRAP	0.914 **	0.631 **	−0.059	0.883 **	0.548 *	1.000

* Significantly correlated at 0.05 level; ** extremely correlated at 0.01 level.

**Table 4 molecules-27-05185-t004:** Scientific classification of six citrus cultivars studied in this study.

Varieties	Common Name	Species Name	Abbreviation
Huyou	Grapefruit	Citrus paradisi Macf. Changshanhuyou	HY
Tiancheng	Sweet orange	Citrus sinensis Osbeck	TC
Zaoshuwenzhoumigan	Mandarin	Citrus unshiu var. praecox Tanka cv Miyagawa wase	ZS
Wanshuwenzhoumigan	Mandarin	Citrus unshiu Marc. cv Yamada	WS
Ougan	Mandarin	Citrus suavissima Hort. ex Tanaka	OG
Ponkan	Mandarin	Citrus poonensis Hort. ex Tanaka	PG

## Data Availability

The main supporting data are included within the article.

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
