# Peer review of "Phytochemicals and Antioxidant Capacities of Young Citrus Fruits Cultivated in China"

_molecules, 2022, doi:10.3390/molecules27165185_

Round 1
Reviewer 1 Report
Abstract
- line 17: six varieties of young citrus cultivated in China…
-line 20: indicated that Ougan variety…
-line 21: Wanshuwenzhoumigan variety...
-Lines 23-24: abundant flavones in Ponkan, Ougan, and Wanshuwenzhoumigan varieties…
-Line 26: The young fruits of Ougan and Huyou varieties..
Material
Line 187: How were the young fruits used? your peels? or all the fruit? Exocarp or mesocarp?
Line 188: what were the pre-harvest parameters used (period of collect? local temperature – day or night), altitude, soils type and depth; storage conditions (temperature, humidity, light intensity? agrochemicals? Line 190: what is the amount of fruits used initially? Results 2.1. Total phenolic, total flavonoid, and synephrine contents (Figure 1) -line 80: on the axial axis of Figure-1A place “mg GAE / g DW”
-identify abbreviations for the fruit varieties of the Figure 1 in the text;
-line 95: identify abbreviations for the fruit varieties in table 1;
Line 112: identify abbreviations for the fruit varieties in table 2;
Line 144: identify abbreviations for the fruit varieties in figure 2;
DPPH is expressed in “µmolL/g DW”,
but in the method section (4.6.3. DPPH assay)
“…Results were expressed as “mM of TE per g of DW”;
-Line 196: Table 4: in the text the authors use the term ponkan, but in table 4 this term does not appear.
Material and Methods
Line 187: How were the young fruits used? your peels? or all the fruit? Exocarp or mesocarp?
Line 188: what were the pre-harvest parameters used (period of collect? local temperature – day or night), altitude, soils type and depth; storage conditions (temperature, humidity, light intensity? agrochemicals? Line 190: what is the amount of fruits used initially?
Line 207: Please state the exact amount of material used in each 4.3.1, 4.3.2, 4.3.3, 4.4., and
4.5 analysis.
FRAP assay Line 272: what appropriate dilution concentration was used? References The authors could include the reference: Antioxidant activity of Citrus fruits
September 2015, Food Chemistry 196, DOI: 10.1016/j.foodchem.2015.09.072
Zhuo Zou, Wanpeng Xi, Yan Hu, Chao Nie, Zhiqin Zhou

Reviewer 2 Report
Manuscript Number: molecules-1858986
Title of the manuscript: Phytochemicals and antioxidant capacities of young citrus fruits cultivated in China
COMMENTS TO THE AUTHOR:
The manuscript "Phytochemicals and antioxidant capacities of young citrus fruits cultivated in China " is scientifically sound and has focused on the identification of phenolics and screening of antioxidant activity study of young citrus fruits. This research will be of high interest to the readers in the understanding potential of citrus fruits phytoconstituents and their importance as antioxidants.
The manuscript can be accepted for publication after the following comments are addressed by the authors
Comments:
Major corrections:
1. Importance of the antioxidant potential of phytoconstituents like phenolics, flavones, synephrine, etc. requires to be included as a separate paragraph in the introduction section
2. The HLPC chromatogram of extracts needs to be added as a figure in the result section.
3. Discussion is very short, and it is just the repetition of methods and results.
A proper discussion requires to be written including information on the significance of antioxidant activity of citrus fruits, the importance of antioxidant effects of phytochemicals that have been analyzed in the study, and how this potential can be used to treat different oxidative stress-induced diseases.
Minor corrections
1. Abstract:
The first sentence can be written as “Fruits of six varieties of young citrus cultivated in China were collected for phytochemical composition analysis and antioxidant activity determination”
Line 18: instead of with HPLC use using HPLC
Grammatical error: Line 24: Antioxidant capacity measured (please add was before measured)
Line 26: Authors have mentioned Huyou as a potential source for extracting phytochemicals along with Ougan but have not written anything about Huyou in the abstract before this sentence to justify their conclusion. In the result section, Huyou was found to contain the second highest phenolic content…
Please fix this issue.
Grammatical and technical errors need to be checked and corrected in the entire manuscript
